# Fuzzy-Based PROMETHEE Method for Performance Ranking of SARS-CoV-2 IgM Antibody Tests

**DOI:** 10.3390/diagnostics12112830

**Published:** 2022-11-17

**Authors:** Ayse Arikan, Tamer Sanlidag, Murat Sayan, Berna Uzun, Dilber Uzun Ozsahin

**Affiliations:** 1DESAM Research Institute, Near East University, TRNC Mersin 10, Nicosia 99138, Turkey; 2Department of Medical Microbiology and Clinical Microbiology, Faculty of Medicine, Near East University, TRNC Mersin 10, Nicosia 99138, Turkey; 3Department of Medical Microbiology and Clinical Microbiology, Kyrenia University, TRNC Mersin 10, Kyrenia 99320, Turkey; 4PCR Unit, Research and Education Hospital, Kocaeli University, Kocaeli 41001, Turkey; 5Department of Statistics, Carlos III Madrid University, 28903 Getafe, Madrid, Spain; 6Department of Mathematics, Near East University, TRNC Mersin 10, Nicosia 99138, Turkey; 7Operational Research Center in Healthcare, Near East University, TRNC Mersin 10, Nicosia 99138, Turkey; 8Department of Medical Diagnostic Imaging, College of Health Sciences, Sharjah University, Sharjah 27272, United Arab Emirates

**Keywords:** SARS-CoV-2, IgM, COVID-19, fuzzy logic, PROMETHEE, MCDM

## Abstract

Antibody tests, widely used as a complementary approach to reverse transcriptase–polymerase chain reaction testing in identifying COVID-19 cases, are used to measure antibodies developed for COVID-19. This study aimed to evaluate the different parameters of the FDA-authorized SARS-CoV-2 IgM antibody tests and to rank them according to their performance levels. In the study, we involved 27 antibody tests, and the analyzes were performed using the fuzzy preference ranking organization method for the enrichment evaluation model, a multi-criteria decision-making model. While criteria such as analytical sensitivity, specificity, positive predictive value, and negative predictive value were evaluated in the study, the ranking was reported by determining the importance levels of the criteria. According to our evaluation, Innovita 2019-nCoV Ab Test (colloidal gold) was at the top of the ranking. While Cellex qSARS-CoV-2 IgG/IgM Rapid Test and Assure COVID-19 IgG/IgM Rapid Tester ranked second and third on the list, the InBios-SCoV 2 Detect Ig M ELISA Rapid Test Kit was determined as the least preferable. The fuzzy preference ranking organization method for enrichment evaluation, which has been applied to many fields, can help decision-makers choose the appropriate antibody test for managing COVID-19 in controlling the global pandemic.

## 1. Introduction

Since its initial detection in December 2019, severe acute respiratory syndrome coronavirus 2 (SARS-CoV-2) has spread rapidly worldwide, causing more than 620 million infections and more than 6.5 million deaths [1]. SARS-CoV-2, belonging to the *Coronaviridae* family, is a single-stranded, positive-sense RNA virus that has exhibited high genetic diversity since its first appearance [2,3]. These changes affect the viral antigenic phenotype and provide a fitness advantage. As a result, emerging SARS-CoV-2 variants may increase the virus transmission rate, leading to hospitalizations and increased mortality rates in all age groups [4]. Therefore, early detection, isolation, and treatment to limit virus transmission play an essential role [5]. 

COVID-19 can be diagnosed by detecting viral nucleic acid either by nucleic acid amplification testing or virus-specific proteins by antigen testing and antibody detection by serology testing [6,7]; detection of viral-specific antibodies can enhance and support the accurate and more precise diagnosis. These tests monitor the progression of the infection and treatment responses to COVID-19 [8,9]. Antibody testing is commonly used to measure the immune response after natural infection and vaccination, predict the duration of immunoglobulin (IgM, IgA, and IgG) responses by infected cases, and in retrospective assessment of the infected population for epidemiological surveillance studies [10,11,12,13]. During the pandemic, different antibody tests have been designed to be used for these purposes. However, the more options, the harder it is to decide. Therefore, new methods are needed to choose the most suitable test for SARS-CoV-2 antibodies.

Multi-criteria decision-making (MCDM) methods began to be developed in the 1960s when several tools were deemed necessary to assist decision-making [14]. The decision-making process will not be accessible in cases where several parameters determine the target to be achieved in the selection. Each of the alternatives to be evaluated for selection has its advantages. In such cases, the decision maker will either eliminate all these indecision problems, whether healthy or unhealthy or reach a doubt in doubt after long and irrational analyses. Using MCDM methods aims to keep the decision-making mechanism under control in cases where the number of alternatives and parameters (criteria) is high and obtain the decision result as rationally and quickly as possible [14]. Mostly, in complex decision-making problems, there are incomparable and immeasurable situations between the parameters of the alternatives. MCDM methods consider these situations and assist the decision maker (DM) in finding an optimum solution and ranking the other options. Fuzzy-based MCDM methods offer different approaches to DM when there is a situation of incommensurability and incomparability between the alternatives and solutions as opposed to the elimination of these parameters [14]. In real-world problems, while one option is superior to another in one criterion, it is generally not the most ideal in another bar. MCDM methods help DMs with various techniques based on the features of the data of such scenarios or cases.

Many MCDM techniques, such as The Technique for Order of Preference by Similarity to Ideal Solution (TOPSIS), Analytic Hierarchy Process (AHP), Elimination and Choice Translating Reality (ELECTRE), preference ranking organization method for enrichment evaluation (PROMETHEE), Fuzzy-based MCDM techniques., are proposed by different researchers with different advantages and disadvantages. The most important difference and benefit of the PROMETHEE technique are that it provides the decision-makers with varying functions of preference in calculating the preference values of the alternatives for each criterion [14]. 

PROMETHEE method, one of the most recently developed methods among MCDM methods, was improved by Jean-Pierre Brans and Philippe Vincke in 1985 after being introduced by Jean-Pierre Brans in 1982. The key characteristics of the PROMETHEE method are simplicity, clarity, and balance [15]. The technique uses preference functions when creating an order. All parameters must be clearly defined for the analysis. With the PROMETHEE method, it is possible to perform both partial rankings (PROMETHEE I). 

Net ranking (PROMETHEE II) on a finite number of alternatives and the criterion is a result of the main superiority of standards: sensitivity, specificity, positive and negative predictive values (PPV/NPV) as well as is a result of scales the method is a well-constructed decision support system that enables evaluation and decision making based on many criteria benefits of scales. The advantage of scales in this decision matrix is the origin of the PROMETHEE method, as it is for other analytical MCDM methods [15]. In this matrix, alternatives are assessed with different parameters. The application of the PROMETHEE method requires two additional types of information. The first of these is the determination of the weight of the parameters. This criterion’s weight is the standard’s relative importance [15]. The second is the determination of the preference function. The decision maker uses this information to compare the contribution of each criterion to alternatives. The concept of enrichment, mentioned in the exact name of the method, is because the technique is not carried out with a simple process when evaluating the initial decision matrix but is based on previously determined preference functions. The PROMETHEE method provides a more detailed analysis by evaluating alternatives based on different preference functions, providing both partial and net priorities of the option [15]. Sorting by PROMETHEE method has two critical advantages and other sorting methods. The first of these advantages is that a different preference function can be used for each criterion to evaluate the alternatives. The second is that partial and complete rankings of the other options can be obtained. With these advantages, the efficiency and accuracy of the process have been increased in the organizations where the application is made [15]. 

The PROMETHEE technique has handled a significant number of successful applications in a variety of industries, including banking, industrial location, workforce planning, water resources, investments, medicine, chemistry, health care, tourism, ethics in OR, and dynamic management, due to its viability in outranking alternatives and the availability of many versions [14,15]. The methodology’s success is mainly attributable to its mathematical characteristics and unique user-friendliness [14,15]. It has not been applied before for evaluating SARS-CoV-2 IgM antibody test options. 

This study aimed to propose whether the analytical MCDM methods, precisely the fuzzy PROMETHEE approach in this study, which has been used in various fields of health, can guide the selection of diagnostic tests for infectious diseases. As the IgM is the first antibody in response to COVID-19 and is widely used for the diagnosis of acute infection, in this study, we preferred to evaluate the comparative diagnostic performance of the SARS-CoV-2 IgM antibody test kits, considering their parameters, including analytical sensitivity, specificity, positive predictive value, negative predictive value, etc. simultaneously, which is not an easy task even for the experts since many criteria have an impact on the performance of the diagnostic test kits For this purpose, we aimed to rank the performance of the SARS-CoV-2 IgM antibody test kits that have been approved by the Food and Drug Administration (FDA) for emergency use in the diagnosis of acute COVID-19 by evaluation of the standard criteria with fuzzy PROMETHEE, which is pairwise analytical MCDM model successfully applied in many areas where the selection problems arise under the vague environment. With this model, the strengths and the weaknesses of the antibody tests were also analyzed in detail. 

## 2. Materials and Methods

Fuzzy logic-based MCDM techniques are to be used to clarify the uncertainty of the parameters of the selection problems to provide the ideal solutions. MCDM strategies have been extended to a wide range of engineering applications, specifically for materials selection problems, but they are rarely used in medical situations where the complexity is higher. These techniques include analytical and non-analytical approaches based on the nature of the problem, and it aims to reduce the complexity of the problem into a manageable form. Fuzzy logic has been defined as a process of multivalued logic and enables the DM to integrate the non-crisp parameters into systems for consideration. The main components of a decision process are the aim of the problem, which should be well determined, available alternatives, and the parameters/criteria assigned to the other options, as shown in Figure 1. 

The decision-making process generally contains seven steps, as seen in Figure 2. If the solution does not fulfill the decision maker’s needs, the problem structure should be revised, and the process should be re-followed until the ideal solution is obtained.

To construct the decision matrix of the SARS-CoV-2 Ig M antibody tests, we have selected the parameters that can affect their performance based on the expert’s opinion. A total of 27 SARS-CoV-2 IgM antibody tests were involved in this study. These tests have been authorized for emergency use by the FDA, and they were involved in this study based on the information provided by manufacturers’ instructions for use on FDA’s official web page [16]. These tests were not implemented in clinical samples. A new approach (F-PROMETHEE method) interpreted the criteria for these tests to decide the most appropriate IgM diagnostic kit for SARS-CoV-2. Therefore, positive and negative controls were not used as the study did not involve clinical samples. The results were obtained based on the selected parameters and the priorities given to these standards of the experts. The study was conducted by evaluating different variables of SARS-CoV-2 Ig M antibody tests (*n* = 27), including (*n* = 17, 63%) lateral flow assay (LFA) (rapid diagnostic tests), plate-based enzyme-linked immunosorbent assay (*n* = 1, 4%), chemiluminescence immunoassay (CLIA) (*n* = 5, 19%), chemiluminescence microparticle immunoassay (CMIA) (*n* = 3, 11%), and enzyme-linked immunofluorescent assay (ELISA) (*n* = 1, 4%) with F-PROMETHEE method of MCDM technique. 

The criteria provided by the manufacturer’s instructions for use and involvement in the study were analytical sensitivity, specificity, PPV, NPV, sample type and volume, assay technique, antigen target, time to the first result, duration of post-infection sampling days, reagent storage conditions, applicability, accessibility to kits, result storage, capacity, loading capacity/per run, maximum efficiency, frequency of calibration, etc. With the fuzzy PROMETHEE technique, based on the simultaneous interpretation of a large number of criteria/parameters on a model, 27 different SARS-CoV-2 IgM test kits were assessed. This study used fuzzy triangular sets (see Figure 3) to determine the parameters of the alternatives. 

The importance levels of each standard have been assigned using the fuzzy triangular scale based on the experts’ opinions, as shown in Table 1. The scores for the standards preferences for the selected criteria were determined by the experts’ opinions and defined numerically with a triangular fuzzy linguistic scale that ranges between 0 and 1 on the model. As parameters such as sensitivity, specificity, PPV, and NPV are critical to the accuracy of a diagnostic test kit, their importance weights were rated very high in the model, with a fuzzy score of (0.75, 1, 1). Critical criteria were scored with lower fuzzy scores (0.5, 0.75, 1), and the medium and low importance fuzzy scores were assigned as (0.25, 0.50, 0.75) and (0, 0.25, 0.5), respectively. In the scoring, the SARS-CoV-2 pandemic and future pandemics were mainly considered.

After determining the parameter’s importance levels, the Yager index was used for the defuzzification of the given triangular fuzzy values of the linguistic scale. Then, the F-PROMETHEE technique was performed with Gaussian preference functions for each criterion since it gives preference values to alternatives using the standard deviation of the related measures.

## 3. Results 

The IgM tests were ranked according to each test’s net ranking flow value (Phi). Phi of each test was calculated by subtracting negative outranking flow (Phi−) from positive outranking flow (Phi+). Phi+ is a value that represents the strengths of the alternatives. In contrast, Phi− is a value that shows the weaknesses of the options when compared with other choices concerning each criterion and the given importance weights. The data obtained from the interpretation of different parameters show that the Innovita 2019-nCoV Ab Test (colloidal gold) (Innovita Biological Technology Co. Ltd., Tangshan, China) represents the expected test performance best among all SARS-CoV-2 IgM tests. Complete ranking of FDA EMU-authorized SARS-CoV-2 IgM antibody tests in Table 1 results from the main superiority of criteria: sensitivity, specificity, PPV/NPV, and time of sampling days post-infection. According to this ranking, the Cellex qSARS-CoV-2 IgG/IgM Rapid Test (Cardinal Health, Charlotte, NC, USA) and Assure COVID-19 IgG/IgM Rapid Tester (Assure Tech (Hangzhou) Co. Ltd., Hangzhou, China) were the second and third best-performing test kits, respectively. In this study, InBios-SCoV 2 Detect IgM ELISA Rapid Test Kit (InBios International, Inc., Seattle, WA, USA) was considered the least preferred one (see Table 2).

Innovita 2019-nCoV Ab Test (colloidal gold) (Innovita Biological Technology Co. Ltd., Tangshan, China), the most appropriate antigen test according to the ranking, is an LFA system that gives results within 10–15 min. A minimal amount of human serum, plasma, or venous blood obtained ≥8 days after symptom onset is of great advantage for this kit. The sensitivity, specificity, PPV, and NPV values are given as 100%, 97.5%, 97.6%, and 100%, respectively, according to manufacturer instructions. In addition to these criteria, its properties, such as targeting nucleocapsid and spike (S) proteins and being stored at room temperature, were the first in the rank [16].

In addition, the plate-based InBios-SCoV 2 Detect IgM ELISA Rapid Test Kit (InBios International, Inc., Seattle, WA, USA) with a sensitivity of 96.7%, specificity of 98.8%, PPV and NPV values of 98.7% and 96.7%, respectively, has been listed at the end of the list due to these criteria as well as the parameters such as targeting only S protein, providing results in approximately two h with either serum or plasma specimens.

## 4. Discussion

Since the beginning of the pandemic, there has been an effort to manage and control COVID-19 worldwide. As a complement to rRT-PCR, reliable, high-quality serological test kits are crucial in combating the COVID-19 pandemic. Due to the lockdown of many businesses and restrictions on travel, the world economy has been significantly affected. Thus, it is precious to produce different diagnostic assays that provide reliable and rapid results regarding the SARS-CoV-2 infection and immune response in the host against the virus in a short timeframe to prevent future infections, enhance cure rate, prevent deaths, and normalize life. SARS-CoV-2 IgM antibody testing with rRT-PCR can provide a more accurate and precise diagnosis of acute COVID-19 cases.

Recent studies also highlight serological diagnostic tests’ solid and high specificity that supports molecular methods in diagnosing COVID-19 [15,16,17,18]. Accordingly, it may be appropriate to use antibody tests during the pandemic periods. This study ranked the SARS-CoV-2 IgM antibody tests using the fuzzy-based PROMETHEE technique. Here, our findings demonstrated that this new technique, which has been used in SARS-CoV-2 diagnostic approaches, therapeutic options, and potential vaccines before, could be effectively applied to the interpretation of commercially available SARS-CoV-2 IgM antibody tests during global pandemic management [15,19,20]. The F-PROMETHEE method can guide decision-makers in deciding on the most appropriate SARS-CoV-2 IgM antibody test for each country to support their RT-PCR results. The performance ranking of IgM antibody tests revealed that LFA IgM antibody tests are preferable. Rapid diagnostic kits provided the best analytical success in this study and were a point of care for testing in the community.

During a pandemic, timely availability and fast access to reliable, high-quality serologic test techniques as a complement to rRT-PCR would play a tremendous role in fighting pandemics. Various antigen targets, including recombinant full S (*spike*), N (nucleocapsid) proteins, or peptides of the N and S1, S2, and receptor binding domain (RBD) of S protein are used in different SARS-CoV-2 serological test platforms. S and N proteins are the most immunogenic [13]. Recently, The U.S. FDA has allowed various anti-SARS-CoV-2 systems for emergency use to detect antibodies against SARS-CoV-2 [11]. Main commercially available serological tests involve techniques such as LFA, ELISA, CLIA, CMIA, enzyme-linked fluorescent assay (ELFA), photonic ring immunoassay, fluorescent immunoassay (FIA), and fluorescent multiplex bead-based immunoassay (FMIA) have received emergency use authorization for the detection of viral-specific antibodies, which generally develop several days after the first exposure to the virus [11,21]. Throughout the COVID-19 pandemic, there was a need to develop rapid and efficient new methods to diagnose and monitor COVID-19 cases, besides conventional techniques, to limit the virus’s spread. New systems have been proposed to fight against COVID-19 and other infectious diseases. A novel solution is the 5G-enabled ultra-sensitive fluorescence sensor which suggests quantitative detection of SARS-CoV-2 antigens using mesoporous silica encapsulated op conversion nanoparticles labeled as LFA [22]. Therefore, continuous evaluation of the performance of various systems is also required to determine the most appropriate and accurate methods. 

During a pandemic, criteria such as point-of-care testing (POCTs), high return, short turnaround times, accessibility, applicability, and storage conditions of kits also assume an essential role in successful crisis management [23,24,25]. Although the current study allowed us to evaluate the diagnostic performance of all available techniques with different criteria developed for SARS-CoV-2 IgM, recent studies for this purpose have considered only a few methods, either by test technique or antigen target using clinical data [26,27,28,29]. Among those, POCT systems are more popular than other laboratory-based automatized systems. One of the recent studies on rapid tests revealed 100% sensitivity with no cross-reactivity with any common cold agents in SARS-CoV-2 infected cases [23]. Similarly, with the automated systems, Lau et al. assessed the performance of point-of-care systems (POCTs) against SARS-CoV-2. The study revealed that the POCTs had acceptable specificity with little cross-reactivity with other antibodies [30]. 

On the other hand, the method used in this study allows decision-makers to evaluate multiple criteria simultaneously in different IgM antibody test kits and decide on the most preferred technique or equipment to reduce the risk of further virus transmission throughout the pandemic. MCDM methods have been widely used in different fields to remove ambiguity in the selection of alternatives before [14,15,20]. With the COVID-19 pandemic, MCDM methods have been used to evaluate newly developed approaches used in the diagnosis, treatment, and prevention of COVID-19 [17,21,22]. However, this is the first study in the literature that suggests the MCDM (fuzzy PROMETHEE) technique for evaluating the performances and effectiveness of the SARS-CoV-2 IgM antibody tests numerically. Therefore, this method will also guide the different diagnostic test kit alternatives to manage future pandemics effectively. 

## 5. Conclusions

As a complement to rRT-PCR, reliable, high-quality, easy-to-access serological test kits actively combat the COVID-19 epidemic. SARS-CoV-2 IgM antibody testing combined with molecular systems can provide a more accurate and definitive diagnosis of acute COVID-19 cases. Since evaluating conflicting criteria in deciding on the most accurate and appropriate diagnostic tests is impractical, the F-PROMETHEE method can be effectively applied to this field. The technique can guide decision-makers in selecting the most appropriate kits for the diagnosis of COVID-19 distributed or developed in each country. The decision maker can evaluate and compare conflicting criteria based on this technique to determine the most favorable antibody test according to local prevalence for effective COVID-19 management. 

The following are the study’s limitations: (1) due to the lack of studies in the literature on the evaluation of antibody tests by mathematical analysis; likely, the tests included in the study were not widely used during the pandemic period; (2) positive and negative control samples could not be used because the study was based on mathematical data analysis and was not conducted with clinical samples. Based on the nature of the analytical decision-making process, the results were obtained based on the selected parameters and the experts’ preferences for determining the importance of the criteria, which can be updated based on the decision-makers priorities. Furthermore, the results provided sufficient and supportive information about the SARS-CoV-2 IgM antibody tests since the analysis was carried out by considering the most critical parameters that have a significant impact on the performances of the SARS-CoV-2 IgM antibody tests.

The study clarifies that different types of diagnostics, treatments, and vaccines developed against emerging and re-emerging infectious agents can also be evaluated with these new approaches in the future. Especially at the beginning of future outbreaks, the performance of newly developed diagnostic and therapeutic strategies can be assessed using medical data with mathematical models to guide decision-makers in the clinical management of infectious diseases. Mathematical analysis of medical data will be an impressive solution to predict future outbreaks and their measurements.

## Figures and Tables

**Figure 1 diagnostics-12-02830-f001:**
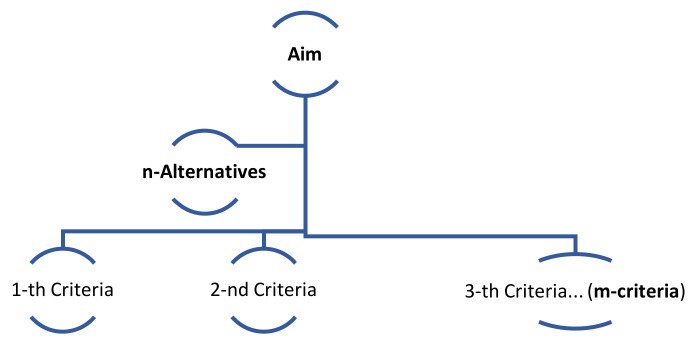
The general form of the decision-making analysis components.

**Figure 2 diagnostics-12-02830-f002:**
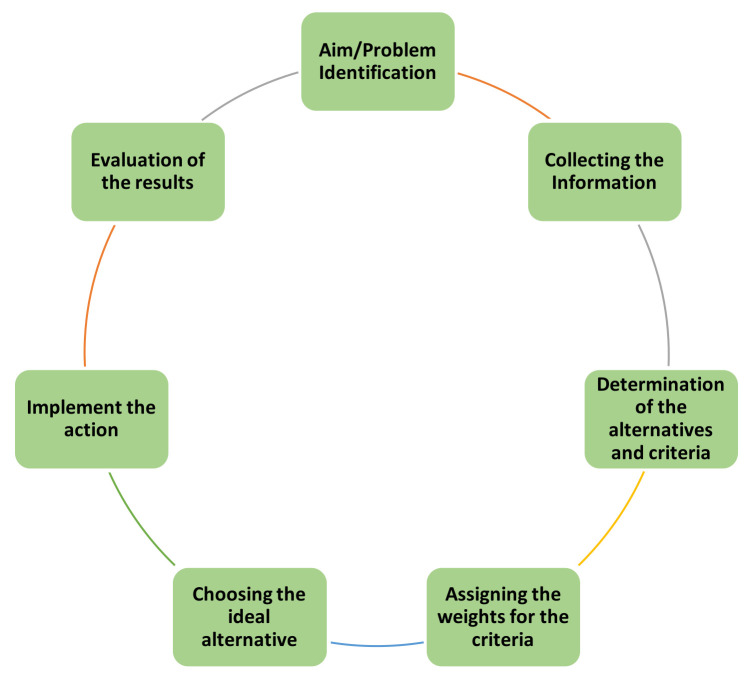
The process of the decision-making applications.

**Figure 3 diagnostics-12-02830-f003:**
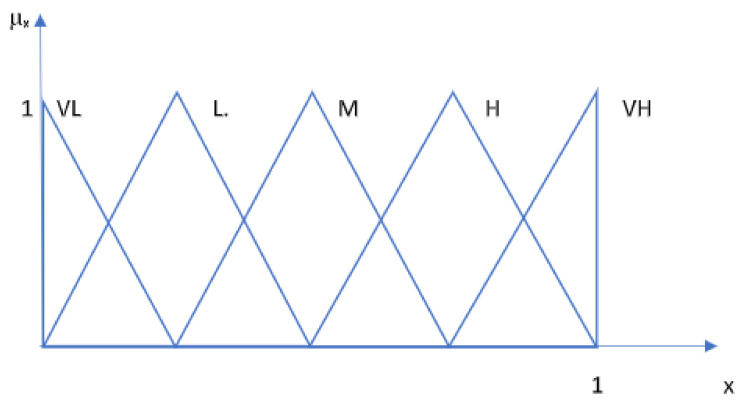
The triangular linguistic sets were used for the fuzzification process of the dataset. (Very high: VH, high: H, medium: M, low: L, very low: VL, μ_x_: membership values of the x values assigned to linguistic classes).

**Table 1 diagnostics-12-02830-t001:** Fuzzy linguistic scale and the importance levels of the selected parameters.

Linguistic/Triangular Fuzzy Scale	Criteria/Parameter
(Very high)/(0.75, 1, 1)	Sensitivity IgM, specificity IgM, positive predictive value, negative predictive value, calibration frequency
(High)/(0.50, 0.75, 1)	Target, result time, result interpretation, storage, time of sampling days post symptom onset, maximum throughput/hour
(Moderate)/0.25, 0.50, 0.75)	Specimen type, technology, practicability/ease of operation, loading capacity/each run, access to kit, result storage, special laboratory equipment requirement
(Low)/(0, 0.25, 0.50)	Sample volume, test kit size
(Very low)/(0, 0, 0.25)	None of the criteria

**Table 2 diagnostics-12-02830-t002:** Complete ranking of FDA EUAs SARS-CoV-2 IgM antibody tests.

Ranking	SARS-CoV-2 IgM Diagnostic Test Kit	Test Technique	Phi	Phi+	Phi−
1	Innovita	CLIA	0.0270	0.0348	0.0078
2	Cellex	LFA	0.0241	0.0335	0.0094
3	Assure	LFA	0.0240	0.0335	0.0095
4	ACON	LFA	0.0234	0.0335	0.0101
5	Nirmidas MidaSpot	LFA	0.0200	0.0306	0.0106
6	Salofa Sienna-Clarity	LFA	0.0200	0.0306	0.0106
7	Biocan	LFA	0.0193	0.0292	0.0100
8	Hangzhou RightSign	LFA	0.0152	0.0263	0.0111
9	Healgen	LFA	0.0152	0.0263	0.0111
10	Xiamen BIOTIME	LFA	0.0152	0.0263	0.0111
11	Hangzhou LYHER	Colloidal gold LFA	0.0145	0.0259	0.0114
12	Megna	LFA	0.0143	0.0304	0.0160
13	Nirmidas	LFA	0.0111	0.0243	0.0132
14	Biohit	Colloidal gold LFA	0.0095	0.0260	0.0165
15	Access	LFA	0.0063	0.0312	0.0249
16	DiaSorin LIAISON	CMIA	−0.0040	0.0302	0.0343
17	Jiangu Orawell	LFA	−0.0116	0.0155	0.0271
18	TBG	LFA	−0.0119	0.0144	0.0263
19	Abbott Advise DX (Alinity)	CMIA	−0.0153	0.0322	0.0475
20	bioMerieux VIDAS	ELFA	−0.0155	0.0167	0.0321
21	Shenzhen MAGLUMI	CLIA	−0.0166	0.0329	0.0495
22	Abbott AdviseDx (Architect)	CMIA	−0.0180	0.0313	0.0493
23	Beckman Coulter Access	CLIA	−0.0192	0.0234	0.0426
24	Diazym DZ Lite	CLIA	−0.0214	0.0320	0.0534
25	BioCheck IgM	CLIA	−0.0227	0.0145	0.0371
26	BioCheck IgG/IgM Combo	CLIA	−0.0252	0.0167	0.0418
27	InBios	ELISA	−0.0658	0.0143	0.0801

Abbreviations: CLIA: chemiluminescence immunoassay; LFA: lateral flow assay; CMIA: chemiluminescent microparticle immunoassay; ELFA: enzyme-linked fluorescence assay; Phi: net ranking flow; Phi+: positive outranking flow, Phi−: negative outranking flow. Phi+ is a value that represents the strengths of the alternatives. In contrast, Phi− is a value that shows the weaknesses of the options when compared with other possibilities concerning each criterion and the given importance weights.

## Data Availability

The datasets generated during and analysed during the current study are available from the corresponding author on reasonable request.

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
