# Peer review of "Fuzzy-Based PROMETHEE Method for Performance Ranking of SARS-CoV-2 IgM Antibody Tests"

_diagnostics, 2022, doi:10.3390/diagnostics12112830_

Round 1
Reviewer 1 Report
The manuscript demonstrated a fuzzy based PROMETHEE method for COVID-19 antibody test. Overall, it is well organized but needs some necessary revisions before publication.
1. Antigen diagnosis of SARS-CoV-2 based on Up-conversion LFA is suggested to be included. For example, Biosensors and Bioelectronics 181 (2021) 113160 "5G-enabled ultra-sensitive fluorescence sensor for proactive prognosis of COVID-19"
2. Figure 2 needs to be refined.
3. How was the performance of PROMETHEE technique in antibody test compared with prior related works?
4. The motivation of this study needs to be elaborated more in detail.
5. The limitations also should be included in the Conclusion part.
Author Response
We are grateful to the reviewers for their positive comments and the time they spent on this study.
Please see the point by point our answers below:
1.Antigen diagnosis of SARS-CoV-2 based on Up-conversion LFA is suggested to be included. For example, Biosensors and Bioelectronics 181 (2021) 113160 "5G-enabled ultra-sensitive fluorescence sensor for proactive prognosis of COVID-19"
Antigen diagnosis of SARS-CoV-2 based on Up-conversion LFA has been added to the main text and highlighted in red, as shown below.
“During the COVID-19 pandemic, there was an urgent need to develop rapid and efficient new methods to diagnose and monitor COVID-19 cases and limit the spread of the virus. Besides conventional techniques, new systems have been proposed to fight against COVID-19 and other infectious diseases. 5G-enabled ultra-sensitive fluorescence sensor is one solution that proposes quantitative detection of SARS-CoV-2 antigens by using mesoporous silica encapsulated op conversion nanoparticles labeled LFA (Guo L 2021).”
2.Figure 2 needs to be refined.
It has been refined.
3.How was the performance of PROMETHEE technique in antibody test compared with prior related works?
The PROMETHEE technique has handled a significant number of successful applications in a variety of industries, including banking, industrial location, manpower planning, water resources, investments, medicine, chemistry, health care, tourism, ethics in OR, dynamic management, etc., due to its viability in outranking alternatives and the availability of many versions [14,15]. The methodology's success is mainly attributable to its mathematical characteristics and unique user-friendliness [14,15]. It has not been applied before for evaluating SARS-CoV-2 IgM antibody test options.
‘Although the current study allowed us to evaluate all available techniques with different antigen targets developed for SARS-CoV-2 IgM, recent studies for this purpose have evaluated the performance of only a few techniques with different antigen targets simultaneously using clinical data’
4.The motivation of this study needs to be elaborated more in detail.
The aim of the study has been clarified and highlighted in red in the main text below:
Added to the introduction section:
“This study aimed to propose whether the analytical MCDM methods, precisely the fuzzy PROMETHEE approach in this study, which has been used in various fields of health, can guide the selection of diagnostic tests for infectious diseases. As the IgM is the first antibody in response to COVID-19 and is widely used for the diagnosis of acute infection, in this study, we preferred to evaluate the comparative diagnostic performance of the SARS-CoV-2 IgM antibody test kits, considering their parameters including analytical sensitivity, specificity, positive predictive value, negative predictive value, etc. simultaneously, which is not an easy task even for the experts since many criteria have an impact on the performance of the diagnostic test kits For this purpose, we aimed to rank the performance of the SARS-CoV-2 IgM antibody test kits that have been approved by the Food and Drug Administration (FDA) for emergency use in the diagnosis of acute COVID-19 by evaluation of the standard criteria with fuzzy PROMETHEE, which is pairwise analytical MCDM model successfully applied in many areas where the selection problems arise under the vague environment. With this model, the strengths and the weaknesses of the antibody tests were also analyzed in detail’’.
Added to Discussion Section:
“Throughout the COVID-19 pandemic, there was a need to develop rapid and efficient new methods to diagnose and monitor COVID-19 cases, besides conventional techniques, in order to limit the spread of the virus. New systems have been proposed to fight against COVID-19 and other infectious diseases. 5G-enabled ultra-sensitive fluorescence sensor is a novel solution that proposes quantitative detection of SARS-CoV-2 antigens using mesoporous silica encapsulated op conversion nanoparticles labeled LFA (22). Therefore, continuous evaluation of the performance of various systems is required to determine the most appropriate and accurate methods.”
5.The limitations also should be included in the Conclusion part.
The limitation of the study has been added to the conclusion and highlighted in red, as shown below.
‘’To point out the study's limitations, (1) Due to the lack of study in the literature on the evaluation of antibody tests by mathematical analysis, it is likely that the tests included in the study were not widely used during the pandemic period (2) Positive and negative control samples could not be used because the study was based on mathematical data analysis and was not conducted with clinical samples. Based on the nature of the analytical decision-making process, the results were obtained based on the selected parameters and the experts' preferences for determining the importance of the criteria, which can be updated based on the decision-makers' priorities. Furthermore, the results provided sufficient and supportive information about the SARS-CoV-2 IgM antibody tests since the analysis was done by considering the most important parameters that have a great impact on the performances of the SARS-CoV-2 IgM antibody tests.
The study clarifies that different types of diagnostics, treatments, and vaccines developed against emerging and re-emerging infectious agents can also be evaluated with these new approaches in the future. Especially at the beginning of future outbreaks, the performance of newly developed diagnostic and therapeutic strategies can be assessed using medical data with mathematical models to guide decision-makers in the clinical management of infectious diseases. Mathematical analysis of medical data will be an impressive solution to predict future outbreaks and their measurements’’.
Kindest Regards
Reviewer 2 Report
Arikan and Colleagues proposed an interesting evaluation of 27 SARS-CoV-2 IgM antibody tests, ranking them according to their performance levels.
I think that the manuscript is not sounding since both some controls and experimental design are missing.
Major:
All the figures should be edited improving both quality and clarity. Some of them (Figure 1) are very unclear. Table 1 should be reformatted.
How the Authors established true positive and true negative? Control is missed in Materials and Methods section.
I think that the Materials and Methods section should be improved. I’m not able to understand the number of participants of the study, if they were recruited in hospital or other health structures, their number and, most important, if each patient was tested with all the methods. I think that this information are crucial to understood the design of the experiment described in the manuscript.
Author Response
We are grateful to the reviewers for their positive comments and the time they spent on this study.
Please see our point by point answers as below:
- All the figures should be edited improving both quality and clarity. Some of them (Figure 1) are very unclear. Table 1 should be reformatted.
The tables and figures are improved.
2.How the Authors established true positive and true negative? Control is missed in Materials and Methods section.
Positive and negative controls could not be used because the study was based on mathematical data analysis and was not conducted with clinical samples.
3. I think that the Materials and Methods section should be improved. I’m not able to understand the number of participants of the study, if they were recruited in hospital or other health structures, their number and, most important, if each patient was tested with all the methods. I think that this information are crucial to understood the design of the experiment described in the manuscript.
The materials and method section has been revised according to the reviewer’s comments.
“This study aimed to propose whether the analytical MCDM methods, precisely the fuzzy PROMETHEE approach in this study, which has been used in various fields of health, can guide the selection of diagnostic tests for infectious diseases. As the IgM is the first antibody in response to COVID-19 and is widely used for the diagnosis of acute infection, in this study, we preferred to evaluate the comparative diagnostic performance of the SARS-CoV-2 IgM antibody test kits, considering their parameters including analytical sensitivity, specificity, positive predictive value, negative predictive value, etc. simultaneously, which is not an easy task even for the experts since many criteria have an impact on the performance of the diagnostic test kits For this purpose, we aimed to rank the performance of the SARS-CoV-2 IgM antibody test kits that have been approved by the Food and Drug Administration (FDA) for emergency use in the diagnosis of acute COVID-19 by evaluation of the standard criteria with fuzzy PROMETHEE, which is pairwise analytical MCDM model successfully applied in many areas where the selection problems arise under the vague environment. With this model, the strengths and the weaknesses of the antibody tests were also analyzed in detail’’.
added to study to be more clearly understandable.
Kindest Regards
Author Response
We are grateful to the reviewers for their positive comments and the time they spent on this study.
Please see our point by point answers below:
- Minor grammar and syntax issues need correction for enhancing readability.
Relevant arrangements have been made.
- Remove abbreviation from Abstract.
The abbreviation in the abstract has been removed.
- Polish merits to your proposed method and what are limitations of the method.
The study's aim and limitations have been highlighted in red in the main text.
- On page 5, author should revise table and present it more clearly.
The detailed information about the table is added and highlighted in red color.
- On page 5 and 6, align text properly.
It has been rearranged.
- The conclusions should be extended with more future work.
Future work is added to the conclusion.
- Irrelevant references should be deleted from the list of references.
It has been applied
- Cite all references in tex.
We updated the references and rearranged them.
- More related recent references to present study could be included.
We have included and cited recent studies.
Kindest Regards
Round 2
Reviewer 2 Report
Authors modified manuscript according to Reviewers' suggestions. They improved MeM section making the manuscript more readable.
I think that it is suitable for publication after a modification of both Figure 2 (please, remove orange bar and shadows, and correct a typo) and Figure 3 (please, avoid black in figure's background).